# Analyzing Deep Transformer Models for Time Series Forecasting via Manifold Learning

## Abstract

Deep transformer models consistently achieve groundbreaking results on natural language processing and computer vision problems, among other engineering and scientific domains. However, despite active research that aims to better understand transformer neural networks via e.g., computing saliency scores or analyzing their attention matrix, these models are not well-understood at large. This problem is further exacerbated for deep time series forecasting methods, for which analysis and understanding work is relatively scarce. Indeed, deep time series forecasting methods only recently emerged as state-of-the-art, and moreover, time series data may be less "natural" to interpret and analyze, unlike image and text information. Complementary to existing analysis studies, we employ a *manifold learning* viewpoint, i.e., we assume that latent representations of time series forecasting models lie next to a low-dimensional manifold. In this work, we study geometric features of latent data manifolds including their intrinsic dimension and principal curvatures. Our results demonstrate that deep transformer models share a similar geometric behavior across layers, and that geometric features are correlated with model performance. Further, untrained models present different structures, which rapidly converge during training. Our geometric analysis and differentiable tools may be used in designing new and improved deep forecasting neural nets.

## 1 Introduction

Over the past decade, modern deep learning has shown remarkable results on multiple challenging tasks in computer vision (Krizhevsky et al., 2012), natural language processing (NLP) (Pennington et al., 2014), and speech recognition (Graves et al., 2013), among other domains (Goodfellow et al., 2016). Recently, the transformer (Vaswani et al., 2017) has revolutionized NLP by allowing neural networks to capture long-range dependencies and contextual information effectively. In addition, transformer-based architectures have been extended to non-NLP fields, and they are among the state-of-the-art (SOTA) models for vision (Dosovitskiy et al., 2020) as well as time series forecasting (TSF) (Wu et al., 2021; Zhou et al., 2022). Unfortunately, while previous works, e.g., (Zeiler & Fergus, 2014; Karpathy et al., 2015; Tsai et al., 2019) and many other works, attempted to explain the underlying mechanisms of neural networks (including the transformer), deep models are still considered not well understood.

The majority of approaches analyzing the inner workings of vision and NLP transformer models investigate their attention modules (Bahdanau et al., 2015) and salient inputs (Wallace et al., 2019). Unfortunately, time series forecasting methods have received significantly less attention. This may be in part due to their relatively recent appearance as strong contenders on TSF in comparison to non-deep and hybrid techniques (Oreshkin et al., 2020). Further, while vision and NLP modalities may be "natural" to interpret and analyze, time series data requires a different set of analysis tools which may be challenging to develop for deep models. For instance, N-BEATS (Oreshkin et al., 2020) designed a method that promotes the learning of trend and seasonality components, however, their model often recovers latent variables whose relation to trend and seasonality is unclear (Challu et al., 2022). Moreover, there is already a significant body of work of SOTA TSF that warrants analysis and understanding. Toward bridging this gap, we investigate in this work the geometric properties of latent representations of transformer-based TSF techniques via *manifold learning* tools.

Manifold learning is the study of complex data representations under the *manifold hypothesis* where high-dimensional data is assumed to lie close to a low-dimensional manifold (Coifman & Lafon, 2006). This assumption is underlying the development of numerous machine learning techniques, akin to considering independent and identically distributed (i.i.d.) samples (Goodfellow et al., 2016). Recent examples include works on vision (Nguyen et al., 2019), NLP (Hashimoto et al., 2016), and time series forecasting (Papaioannou et al., 2022). However, to the best of our knowledge, there is no systematic work that analyzes transformer-based TSF deep neural networks from a manifold learning perspective. In what follows, we advocate the study of geometric features of Riemannian manifolds (Lee, 2006) including their *intrinsic dimension* (ID) and *mean absolute principal curvature* (MAPC). The ID is the minimal degrees of freedom needed for a lossless encoding of the data, and MAPC measures the deviation of a manifold from being flat.

Previously, geometric features of data manifolds were considered in the context of analyzing deep convolutional neural networks (CNN) (Ansuini et al., 2019; Kaufman & Azencot, 2023). Motivated by these recent works, we extend their analysis on image classification to the time series forecasting setting, focusing on SOTA TSF models (Wu et al., 2021; Zhou et al., 2022) evaluated on several multivariate time series datasets. We aim at characterizing the dimension and curvature profiles of latent representations along layers of deep transformer models. Our study addresses the following questions: (i) how do dimensionality and curvature change across layers? are the resulting profiles similar for different architectures and datasets? (ii) is there a correlation between geometric features of the data manifold to the performance of the model? (iii) how do untrained manifolds differ from trained ones? how do manifolds evolve during training?

Our results indicate that the latent manifolds of deep transformer forecasting models undergo two phases: during encoding, dimensionality and curvature either drop or stay fixed, and then, during the decoding part, both dimensionality and curvature increase with respect to their values at the beginning of the decoder. Further, this behavior is shared across several architectures, datasets and forecast horizons. Indeed, regression models as TSF produce outputs from the same distribution as the inputs, and thus, the decoder is expected to yield a complex manifold, whereas the encoder tries to simplify the latent representation (LeCun et al., 2015) to facilitate forecasting. In addition, we find that the intrinsic dimension is inversely proportional to the test mean squared error, allowing one to compare models without access to the test set. Moreover, this correlation is unlike the one found in deep neural networks for classification, which may shed light into the differences between regression and classification. Essentially, the geometric profile of better-performing TSF models should match the geometry of input data, that may be high-dimensional, and thus, explaining the inverse correlation we found. Finally, untrained models show somewhat random dimension and curvature patterns, and moreover, geometric manifolds converge rapidly (within a few epochs) to their final geometric profiles. This finding may be related to studies on the neural tangent kernel (Li & Liang, 2018; Jacot et al., 2018) and linear models for forecasting (Zeng et al., 2023). We believe that our geometric insights, results and tools may be used to design new deep forecasting tools based on the transformer and on other deep neural networks.

## 2 RELATED WORK

Our research lies at the intersection of understanding deep transformer-based models, and manifold learning for analysis and time series. Thus, our discussion below focuses on these topics.

**Analysis of transformers.** Large transformer models have impacted the field of NLP and have led to works such as Vig (2019) that analyze the multi-head attention patterns and found that specific attention heads can be associated with various grammatical functions, such as co-reference and noun modifiers. Several works (Clark et al., 2019; Tenney et al., 2019; Rogers et al., 2021) study the BERT model (Devlin et al., 2019) and show that lower layers handle lexical and syntactic information such as part of speech, while the upper layers handle increasingly complex information such as semantic roles and coreference. In (Dosovitskiy et al., 2020), the authors inspect patch-based vision transformers (ViT) and find that, globally, the models attend to image regions that are semantically relevant for classification. Caron et al. (2021) show that a self-supervised trained ViT produces explicit representations of the semantic location of objects within natural images. Chefer et al. (2021) compute a relevancy score for self-attention layers that is propagated throughout the network, yielding a visualization that highlights class-specific salient image regions. Nie et al. (2023) recently

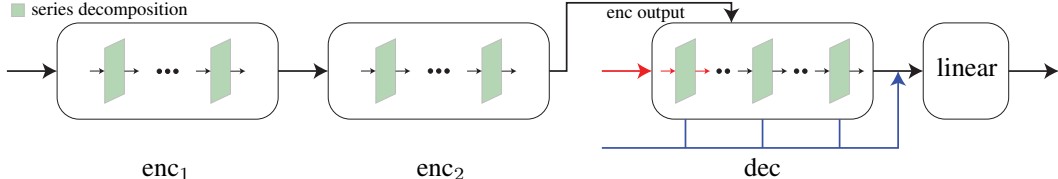

Figure 1: We study Autoformer (Wu et al., 2021) and FEDformer (Zhou et al., 2022) architectures that include two encoders and one decoder, and an output linear layer. We sample geometric features after every sequence decomposition layer, depicted as small rectangles within the blocks.

studied the effectivity of transformer in TSF in terms of their ability to extract temporal relations, role of self-attention, temporal order preservation, embedding strategies, and their dependency on train set size. While in general Zeng et al. (2023) question the effectivity of transformer for forecasting, new transformer-based approaches continue to appear (Nie et al., 2023), consistently improving the state-of-the-art results on common forecasting benchmarks.

**Manifold learning analysis.** Motivated by the ubiquitous manifold hypothesis, several existing approaches investigate geometric features of data representations across different layers. In (Hauser & Ray, 2017), the authors formalize a Riemannian geometry theory of deep neural networks (DNN) and show that residual neural networks are finite difference approximations of dynamical systems. Yu et al. (2018) compare two neural networks by inspecting the Riemann curvature of the learned representations in fully connected layers. Cohen et al. (2020) examine the dimension, radius and capacity throughout the training process, and they suggest that manifolds become linearly separable towards the end of the layer's hierarchy. Doimo et al. (2020) analyzed DNNs trained on ImageNet and found that the probability density of neural representations across different layers exhibits a hierarchical clustering pattern that aligns with the semantic hierarchy of concepts. Stephenson et al. (2021) conclude that data memorization primarily occurs in deeper layers, due to decreasing object manifolds' radius and dimension, and that generalization can be restored by reverting the weights of the final layers to an earlier epoch. Perhaps closest to our approach are the works by (Ansuini et al., 2019) and (Kaufman & Azencot, 2023), where the authors estimate the intrinsic dimension and Riemannian curvature, respectively, of popular deep convolutional neural networks. Both works showed characteristic profiles and a strong correlation between the estimated geometric measure and the generalization error. Recently, Valeriani et al. (2023) investigated the intrinsic dimension and probability density of large transformer models in the context of classification tasks on protein and genetic sequence datasets. Complementary to previous works, our study focuses on the setting of regression time series forecasting problems using multivariate real-world time series datasets.

**Manifold learning in time series forecasting.** Unfortunately, latent representations of deep TSF received less attention in the literature, and thus we discuss works that generally investigate TSF from a manifold learning perspective. Papaioannou et al. (2022) embed high-dimensional time series into a lower-dimensional space using nonlinear manifold learning techniques to improve forecasting. Similarly, Han et al. (2018) proposed a novel framework, which performs nonuniform embedding, dynamical system revealing, and time-series prediction. In Li et al. (2021), the authors exploit manifold learning to extract the low-dimensional intrinsic patterns of electricity loads, to be used as input to recurrent modules for predicting low-dimensional manifolds. Lin et al. (2006) employ a dynamic Bayesian network to learn the underlying nonlinear manifold of time series data, whereas Shnitzer et al. (2017) harness diffusion maps to recover the states of dynamical systems. Finally, distance functions for time series were proposed in (Rodrigues et al., 2018; O'Reilly et al., 2017).

## 3 BACKGROUND AND METHOD

**Time series forecasting.** Given a dataset of multivariate time series sequences $\mathcal{D} := \{x^j_{1:T+h}\}^N_{j=1}$ where $x_{1:T+h} = x_1, \ldots, x_{T+h} \subset \mathbb{R}^D$, the main goal in time series forecasting (TSF) is to accurately forecast the series $x_{T+1:T+h}$, based on the sequence $x_{1:T}$, where we omit $j$ for brevity. The values $T$ and $h$ are typically referred to as lookback and horizon, respectively. The forecast accuracy can be measured in several ways of which the mean squared error (MSE) is the most common. We

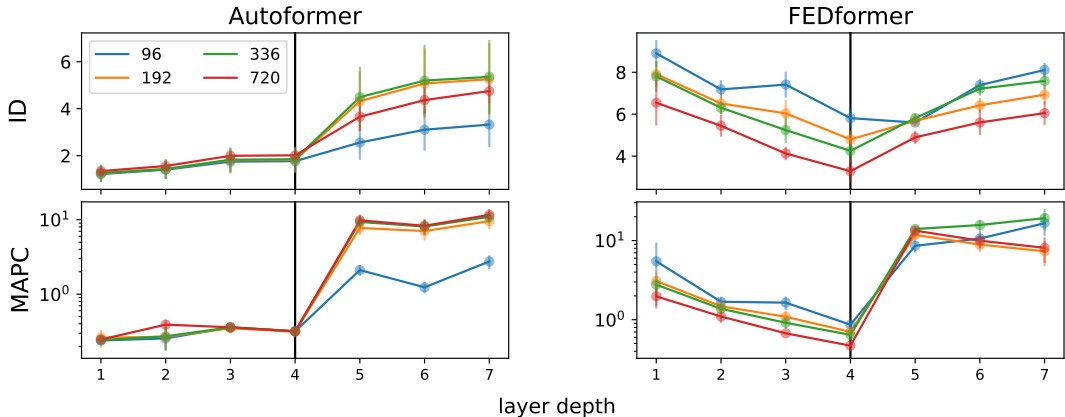

Figure 2: **Intrinsic dimension and mean absolute principal curvature along the layers of Autoformer and FEDformer on traffic dataset for multiple forecasting horizons.** Top) intrinsic dimension. Bottom) mean absolute principal curvature. For each model, both ID and MAPC share a similar profile across different forecasting horizons.

denote by $\tilde{x}_{T+1:T+h} = f(x_{1:T})$ the output of a certain forecast model, e.g., a neural network, then $e_{\mathrm{MSE}} := \frac{1}{h}\sum_{t=T+1}^{T+h} \|x_t - \tilde{x}_t\|_2^2$ is the forecast error. In our study, we consider $T = 96$, and $h = 96, 192, 336$ and $720$, and standard benchmark datasets including Electricity, Traffic, ETTm1, ETTm2, ETTh1, ETTh2, and weather (Wu et al., 2021). In App. A, we provide a detailed description of the datasets and their properties.

**Transformer-based TSF deep neural networks.** State-of-the-art (SOTA) deep time series forecasting models appeared only recently (Oreshkin et al., 2020), enjoying a rapid development of transformer-based architectures, e.g., (Zhou et al., 2021; Wu et al., 2021; Liu et al., 2021; Zhou et al., 2022; Nie et al., 2023), among many others. In what follows, we will focus on Autoformer (Wu et al., 2021) and FEDformer (Zhou et al., 2022) as they are established architectures that are still considered SOTA. In App. C, we also mention additional TSF models and their analysis. Please see Fig. 1 for a schematic illustration of the architecture we investigate. The network is composed of two encoder blocks and a single decoder block, where the encoder and decoder blocks include two and three sequence decomposition layers, respectively. Both Autoformer and FEDformer utilize these decomposition layers to extract trend and seasonality information. In general, the network includes multiple paths from input to output, and here, we focus on the path from the encoder through the decoder to the output. Namely, we discard data from the red and blue trajectories (see Fig. 1), and we use information from the black path. In particular, our analysis is based on sampling geometric properties of the data manifold after every decomposition module and after the final linear layer of the network. We chose the output of the decomposition blocks rather than the attention blocks since the Fourier Cross-correlation layer of the FEDformer model outputs almost identical values for all samples in the series, yielding zero curvature estimates.

**Geometric properties of data manifolds.** The fundamental assumption in our work is that data representations computed across layers of transformer-based models lie on Riemannian manifolds (Lee, 2006). We are interested in computing the intrinsic dimension (ID) and the mean absolute principal curvature (MAPC) of the manifold, following recent work on deep CNNs (Ansuini et al., 2019; Kaufman & Azencot, 2023). We compute the ID using the TwoNN method (Facco et al., 2017) that utilizes the Pareto distribution of the ratio between the distances to the two closest neighbors to estimate the dimension. For the MAPC, we employ the curvature aware manifold learning (CAML) technique (Li, 2018) that parametrizes the manifold via its second-order Taylor expansion, allowing to estimate curvatures via the eigenvalues of local Hessian matrices. We provide additional details regarding the estimation of ID and MAPC in App. D.

**Data collection.** In this study, every architecture is trained on all datasets and horizons, using 10 different seed numbers. For every combination of model, dataset, horizon and seed, we extract the

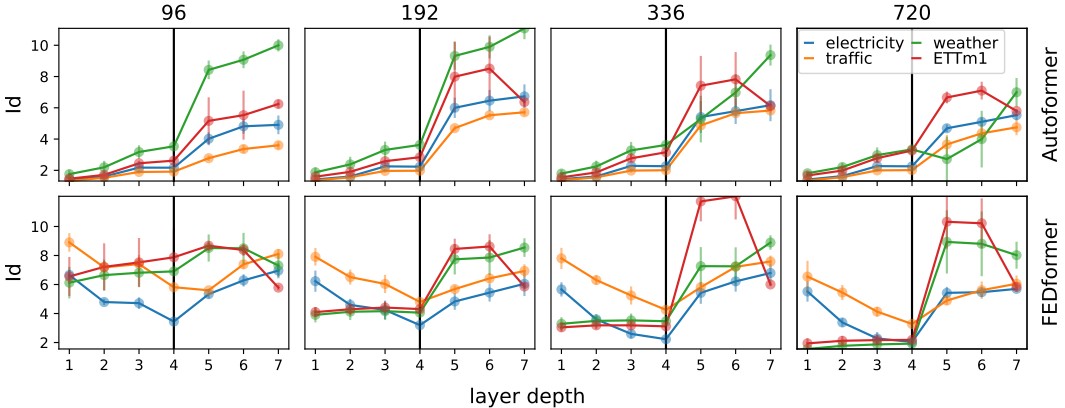

Figure 3: **ID profiles across layers of Autoformer and FEDformer on electricity, traffic, weather and ETTm1 datasets for multiple forecasting horizons.** Each panel includes separate ID profiles per dataset, for several horizons (left to right) and architectures (top to bottom).

latent data representations across layers, and we compute the ID and MAPC. The intrinsic dimension is estimated on 500k point samples from $\mathcal{D}$, resulting in a single scalar value $d$. The estimated ID is used as an input to the CAML algorithm that uses 100k samples, and it returns $d(D - d)$ principal curvatures per point, where $D$ is the extrinsic dimension. To report a single curvature value per manifold, we compute the mean absolute value for each point and we take the mean over all points to obtain the MAPC.

## 4 RESULTS

### 4.1 DATA MANIFOLDS SHARE SIMILAR GEOMETRIC PROFILES

In our first empirical result, we compute the intrinsic dimension (ID) and mean absolute principal curvature (MAPC) across the layers of Autoformer and FEDformer models on the traffic dataset. In Fig. 2, we plot the ID (top row) and MAPC (bottom row) for Autoformer (left column) and FEDformer (right column) on multiple forecast horizons $= 96, 192, 336, 720$. The $x$-labels refer to the layers we sample where labels 1–4 refer to two sequence decomposition layers per encoder block (and thus four in total), labels 5–6 denote the decoder decomposition layers, and label 7 is the linear output layer, see Fig. 1 for the network scheme. A vertical line was added to the figures to illustrate the transition from the encoders to the decoder. Our results indicate that during the encoding phase, the ID and the MAPC are relatively fixed for Autoformer and decrease for FEDformer, and during the decoder module, these values generally increase with depth. Specifically, the ID values change from $\min(\text{ID}) = 1.2$ to $\max(\text{ID}) = 8.1$, showing a relatively small variation across layers. In comparison, the mean absolute principal curvature values present a larger deviation as they range from $\min(\text{MAPC}) = 0.2$ to $\max(\text{MAPC}) = 19.2$.

Remarkably, it can be observed from Fig. 2 that both Autoformer and FEDformer feature similar ID and MAPC profiles in terms of values. Further, a strong similarity in trend can be viewed across different forecast horizons per method. Moreover, Autoformer and FEDformer differ during the encoding phase (layers 1–4), but match quite well during the decoding and output phases (layers 5–7). Our intrinsic dimension estimations stand in contrast to existing results on classification tasks with CNN and transformer architectures, observing a "hunchback" ID profile (Ansuini et al., 2019; Valeriani et al., 2023). That is, prior work found the intrinsic dimension to increase significantly at the first few layers, and then, it presented a sharp decrease with depth. However, deep classification neural networks essentially recover the related low-dimensional data manifold, facilitating a linear separation of classes (Goodfellow et al., 2016), and thus one may expect a low ID toward the final layers of the network. On the other hand, forecast regression models as we study in this work aim to encode the statistical distribution of input data which is typically of a higher dimension due to spurious data variations. Importantly, while our ID profiles do not exhibit the "hunchback" shape identified in (Ansuini et al., 2019), our estimated ID $d$ is significantly smaller than the extrinsic

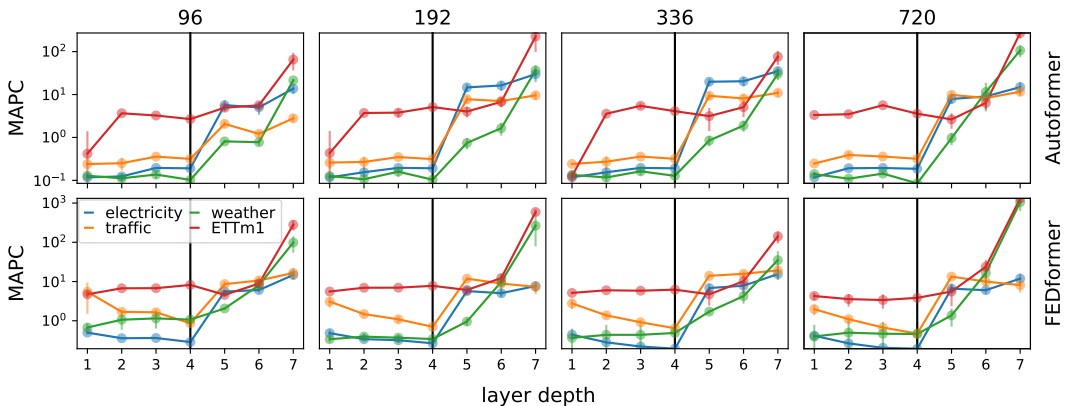

Figure 4: **MAPC profiles across layers of Autoformer and FEDformer on electricity, traffic, weather and ETTm1 datasets for multiple forecasting horizons.** Each panel includes separate MAPC profiles per dataset, for several horizons (left to right) and architectures (top to bottom).

dimension $D = 512$, in correspondence with existing work. Finally, our MAPC profiles attain a "step-like" appearance, similar to the results in (Kaufman & Azencot, 2023), where they identify a sharp increase in curvature in the final layer, and we observe such a jump in the decoder.

To extend our analysis, we present in Fig. 3 and 4 the ID and MAPC profiles, respectively, for Autoformer (top) and FEDformer (bottom) for several horizons using multiple different datasets. For all Autoformer configurations, the IDs in Fig. 3 generally increase with depth, and the IDs of FEDformer present a "v"-shape for electricity and traffic and a "step"-like behavior for weather and ETTm1. Interestingly, ETTm1 (and other ETT* datasets, please see Fig. 8) shows a hunchback trend, however, the drop of ID in the final layer is due to ETT* datasets consisting of a total of seven features, and thus we do not consider this behavior to be characteristic to the network. As in Fig. 2 and existing work (Ansuini et al., 2019), the intrinsic dimension $d$ is much lower than its extrinsic counterpart $D$. Our MAPC results in Fig. 4 indicate a shared step-like behavior in general for all models, horizons, and datasets, where the main difference is where the curvature increase occurs. For electricity and traffic, we observe a sharp increase at the beginning of the decoder block, whereas for weather and ETTm1, the increase often appears at the final layer. Additionally, the maximal curvature values for weather and ETTm1 tend to be higher than those of electricity and traffic. Overall, our results suggest that weather and ETTm1 are associated with manifolds whose geometric features match. This observation can be justified by the known correlation between electricity transformer temperature (ETT) and climate change (Hashmi et al., 2013; Gao et al., 2018). Similarly, electricity consumption (electricity) and road occupancy (traffic) attain a shared behavior that may be explained due to the strong seasonality component in these datasets (Zeng et al., 2023).

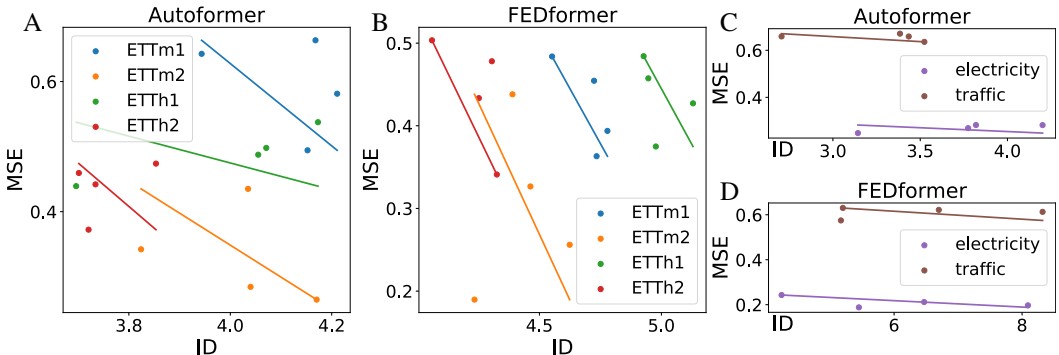

Figure 5: **ID is correlated with model performance.** The test mean squared error is inversely proportional to the intrinsic dimension on ETT* datasets (A, B) and electricity and traffic (C, D).

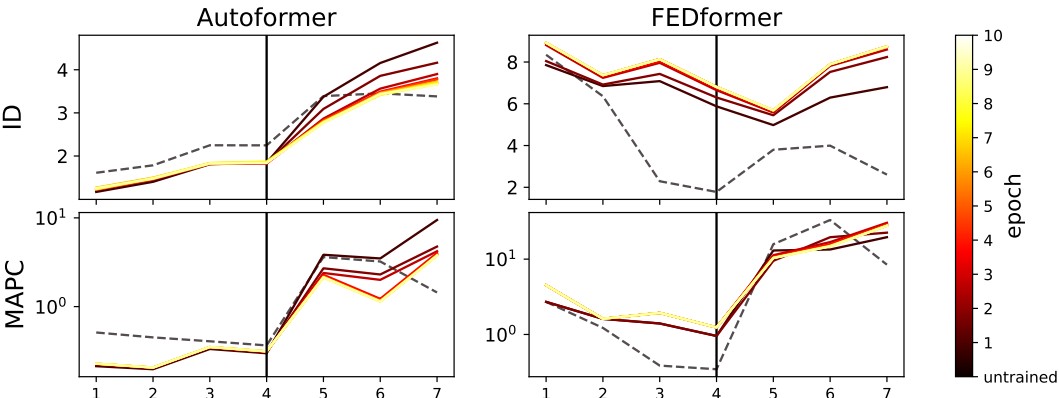

Figure 6: **Training dynamics of the ID and MAPC on traffic dataset.** The plot shows how the ID and MAPC change during training, colored by the training epoch.

## 4.2  FINAL ID IS CORRELATED WITH PERFORMANCE

In what follows, we investigate whether geometric properties of the learned manifold are associated with inherent features of the model. For instance, previous works find a strong correlation between the ID (Ansuini et al., 2019) and MAPC (Kaufman & Azencot, 2023) with model performance and generalization. Specifically, the intrinsic dimension in the last hidden layer is correlated with the top-5 score on image classification, i.e., lower ID is associated with lower error. Similarly, large normalized MAPC gap between the penultimate and final layers of CNNs is related to high classification accuracy. These correlations are important as they allow developers and practitioners to evaluate and compare deep neural networks based on statistics obtained directly from the train set. This is crucial in scenarios where e.g., the test set is unavailable during model design.

We show in Fig. 5 plots of the test mean squared error ($e_{\mathrm{MSE}}$) vs. the intrinsic dimension in the final layer of Autoformer and FEDformer models trained on ETTm1, ETTm2, ETTh1, ETTh2 (panels A and B) and electricity and traffic (panels C and D). For each of the panels, we plot four colored points corresponding to the four different horizons. The colored graphs are generated by plotting the normalized $e_{\mathrm{MSE}}$ with respect to the normalized ID (min-max normalization). We observe a relatively horizontal slope on electricity and traffic, where the ID changes more in comparison to the MSE. On ETT* datasets, we find a negative slope in all Autoformer and FEDformer models. In all cases, we observe an inverse correlation between the test MSE and final ID, namely, the model performs *better* as dimensionality *increases*, where the correlation graphs in Fig. 5A are more approximative in comparison to Fig. 5, panels B, C, D.

As in Sec. 4.1, we identify different characteristics for TSF models with respect to classification neural networks. While popular CNNs show better performance when the intrinsic dimension is lower (Ansuini et al., 2019), we report an opposite trend, namely, better models are associated with a *higher* dimension. Again, this behavior may be attributed to regression networks requiring more degrees of freedom to properly model the statistical distribution of the input information and its large variance. In addition, we note the flat slope profiles presented by electricity and traffic vs. the decreasing curves for ETT* datasets. Essentially, these results indicate that while for ETT* data the manifolds become more expressive in terms of dimensionality and obtaining improved MSEs, transformers yield a relatively fixed MSE on electricity and traffic, regardless of the underlying ID. Our results may hint that Autoformer and FEDformer are *not* expressive enough. Indeed, electricity and traffic include 321 and 862 features, respectively, whereas ETT* datasets have 7 features. Thus, while TSF approaches may need highly expressive networks to model the former datasets due to their complex statistics, it might be that current approaches can not achieve better representations, and they get "stuck" on local minima. We hypothesize that current TSF models are still within the classical ML bias-variance trade-off regime (Goodfellow et al., 2016). In contrast, deep classification models (e.g., He et al. (2016)) exhibit double descent effects (Belkin et al., 2019), forming more expressive and generalizable learning algorithms. We believe that a similar phenomenon of double descent will also emerge for deeper and more expressive TSF models (Nie et al., 2023).

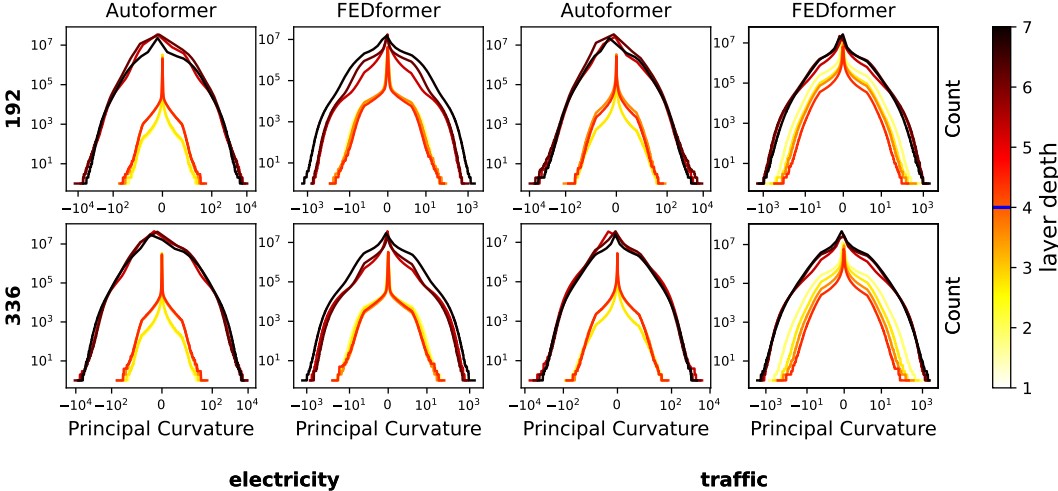

Figure 7: **Principal curvature distribution of Autoformer and FEDformer.** Each plot shows the histogram profiles of principal curvatures per layer, colored by their relative depth. The blue line on the color bar separates the encoders from the decoder.

### 4.3 MANIFOLD DYNAMICS DURING TRAINING AND AT INITIALIZATION

Our analysis above focuses on fully trained deep neural networks and the geometric properties of the learned manifolds. In addition to that analysis, we also investigate below the evolution of manifolds during training and their structure at initialization. In prior works, Ansuini et al. (2019) observed that randomly initialized architectures exhibit a constant ID profile, and further, there is an opposite trend in ID in intermediate layers vs. final layers of convolutional neural networks during training. Kaufman & Azencot (2023) find that untrained models different MAPC profiles than trained networks, and they observe that the normalized MAPC gap consistently increases with model performance during training. Moreover, ID and MAPC profiles converge consistently to their final configuration as training proceeds. Motivated by their analysis, we would like to study the general convergence and trend ID and MAPC of TSF models over several epochs as training evolves.

We show in Fig. 6 the ID and MAPC profiles for Autoformer and FEDformer during training, where each plot is colored by its sampling epoch using the `hot` colormap. First, the untrained ID and MAPC profiles (dashed black) are somewhat random in comparison to the other geometric profiles. Second, the overall convergence to the final behavior is extremely fast, requiring approximately five epochs to converge in all the configurations which is consistent with the results of (Bonheme & Grzes, 2022) where they show that the ID does not change much after the first epoch. Moreover, the encoder in the Autoformer converges within two epochs, whereas the FEDformer model needs more epochs for the encoder to converge. Third, the decoder shows a slower convergence for both methods, suggesting that "most" learning takes place in the decoder component of transformer-based forecasting models. The previous observation aligns with the works of Bonheme & Grzes (2023); Raghu et al. (2017), showing that representations of layers closer to the input tend to stabilize quicker. More specifically, Bonheme & Grzes (2023) show that encoders' representations are generic while decoders' are specific, resulting in a slight change of the encoders' representations during training. Finally, except for the untrained profiles, the ID and MAPC curves during training are generally similar across different epochs. The latter observation may mean that Autoformer and FEDformer mainly perform fine-tuning training as their underlying manifolds do not change much during training.

### 4.4 DISTRIBUTION OF PRINCIPAL CURVATURES

We recall that the CAML algorithm (Li, 2018) we employ for estimating the principal curvatures produces $d(D-d)$ values per point, yielding a massive amount of curvature information for analysis. Following (Kaufman & Azencot, 2023), we compute and plot in Fig. 7 the distribution of principal curvatures for every layer, shown as a smooth histogram for Autoformer and FEDformer models

on electricity and traffic datasets with horizons $= 192, 336$. The histogram plots are colored by the network depth using the `hot` colormap. These distributions strengthen our analysis in Sec. 4.1 where we observe a "step"-like pattern in MAPC, where the sharp jump in curvature occurs at the beginning of the decoder. Indeed, the curves in Fig. 7 related to layers 1–4 span a smaller range in comparison to the curves in layers 5–7. Further, the histograms show that the distribution of curvature is relatively fixed across the encoder blocks, and similarly, a different but rather fixed profile appears in the decoder.

## 5 DISCUSSION

Deep neural networks are composed of several computation layers. Each layer receives inputs from a preceding layer, it applies a (nonlinear) transformation, and it feeds the outputs to a subsequent layer. The overarching theme in this work is the investigation of data representations arising during the computation of deep models. To study these latent representations, we adopt a ubiquitous ansatz, commonly known as the *manifold hypothesis* (Coifman & Lafon, 2006): We assume that while such data may be given in a high-dimensional and complex format, it lies on or next to a low-dimensional manifold. The implications of this inductive bias are paramount; manifolds are rich mathematical objects that are actively studied in theory (Lee, 2006) and practice (Chaudhry et al., 2018), allowing one to harness the abundant classical tools and recent developments to study deep representations.

Our study aligns with the line of works that aims at better understanding the inner mechanisms of deep neural networks. Indeed, while modern machine learning has been dominating many scientific and engineering disciplines since the appearance of AlexNet (Krizhevsky et al., 2012), neural net architectures are still considered not well understood by many. In this context, the manifold ansatz is instrumental—existing analysis works investigate geometric features of latent manifolds and their relation to the underlying task and model performance. For instance, Ansuini et al. (2019) compute the intrinsic dimension of popular convolutional neural networks. Following their work, Kaufman & Azencot (2023) estimate the mean absolute principal curvatures. However, while CNNs are relatively studied from the manifold viewpoint, impactful sequential transformer models (Vaswani et al., 2017), received less attention (Valeriani et al., 2023). The lack of analysis is even more noticeable for the recent state-of-the-art transformer-based *time series forecasting* works, e.g., (Wu et al., 2021). The main objective of our work is to help bridge this gap and study deep forecasting models trained on common challenging datasets from a manifold learning viewpoint.

We compute the intrinsic dimension (ID) and mean absolute principal curvature (MAPC) of data representations from several different deep architectures, forecasting tasks and datasets. To this end, we employ differentiable tools (Facco et al., 2017; Li, 2018) that produce a single scalar ID and many principle curvatures combined to a single scalar MAPC, per manifold. Our results raise several intriguing observations, many of them are in correspondence with existing work. First, the ID is much smaller than the extrinsic dimension, reflecting that learned manifolds are indeed low-dimensional. Second, the ID and MAPC profiles across layers are similar for many different architectures, tasks, and datasets. In particular, we identify two phases, where in the encoder, ID and MAPC are decreasing or stay fixed, and in the decoder, both geometric features increase with depth. Third, the ID in the final layer is strongly correlated with model performance, presenting an inverse correlation, i.e., error is lower when ID is higher. Fourth, we observe that related but different datasets attain similar manifolds, whereas unrelated datasets are associated to manifolds with different characteristics. Finally, untrained models present random ID and MAPC profiles that converge to their final configuration within a few epochs.

Our analysis and observations lie at the heart of the differences between classification and regression tasks; a research avenue that only recently had started to be addressed more frequently (Muthukumar et al., 2021; Yao et al., 2022). Our results indicate a fundamental difference between image classification and time series forecasting models: while the former networks shrink the ID significantly to extract a meaningful representation that is amenable for linear separation, time series forecasting models behave differently. Indeed, the ID generally increases with depth, perhaps to properly capture the large variance of the input domain where regression networks predict. On the other hand, high MAPC seems to be important for classification as well as regression problems. In conclusion, we believe that our work sets the stage for a more general investigation of classification vs. regres-

sion from a manifold learning and other perspectives. We believe that fundamental advancements on this front will lead to powerful machine learning models, better suited for solving the task at hand.

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

## A    TSF DATASETS

Blow we provide a detailed description of the datasets used in the paper. A summery of the datasets can be found in the table A.

**Electricity Transformer Temperature (ETT)(Zhou et al., 2021):**    The ETT contains electricity power load (six features) and oil temperature collected over a period of two years from two countries in China. The dataset is versatile and exhibits short-term periodical patterns, long-term periodical patterns, long-term trends, and irregular patterns. The dataset is further divided to two granularity levels: ETTh1, ETTh2 for one hour level and ETTm1, ETTm2 for 15 minutes level.

**Weather:**[1] The dataset contains 21 meteorological sensors for a range of 1 year in Germany.

**Electricity Consuming Load (ECL):**[2] It contains the hourly electricity consumption (Kwh) of 321 clients.

**Traffic**[3] The dataset consists of hourly data spanning 48 months (2015-2016) obtained from the California Department of Transportation. This data provides information on road occupancy rates, measured by 862 sensors on freeways in the San Francisco Bay area, ranging between 0 and 1.

| Dataset | Number of features | Number of train samples | Granularity |
|---------|--------------------|-----------------------|-------------|
| Ettm1, Ettm2 | 7 | 34369 | 15 minutes |
| Etth1, Etth2 | 7 | 34369 | 1 hour |
| Weather | 21 | 36696 | 1 hour |
| ECL | 321 | 18221 | 1 hour |
| Traffic | 862 | 12089 | 1 hour |

## B    ADDITIONAL RESULTS

### B.1    ETT DATASETS ANALYSIS

To complement the results in the main article we add a comparison of the ID and MAPC of three different ETT datasets: ETTm1, ETTh1 and ETTh2. We notice that ETTm1, ETTh1 and ETTh2 show a hunchback trend for Autoformer while for FEDformer the hunchback trend appears for larger forecast horizons as shown in Fig. 8. Our MAPC results in Fig. 9 show that MAPC are relatively fixed and start to rise at the decoder. Moreover, based on Fig. 3, ETTm1 presents similar ID profiles to electricity and traffic in Autoformer and weather in FEDformer. The main qualitative difference between the ID of ETTm1 and the other datasets is the typical drop in ID in the last layer. This phenomenon can be fully explained by the number of features in ETTm1: the output is of dimension seven, and thus its ID can not be larger than that number. The MAPC results in Fig. 5 show that ETTm1 follows a similar decoding pattern as weather on both Autoformer and FEDformer. However, in Autoformer, ETTm1 exhibits an increase in curvature early on in the encoder. Combining this finding with Fig. 9, we see that other variants of ETT show high MAPC values in Autoformer from the beginning of the encoding phase. A similar behavior appears for the rest of the datasets that present a relatively constant MAPC during encoding.

### B.2    ARCHITECTURE VARIATIONS

This subsection includes a supplementary analysis of the Autoformer and FEDfromer models with varying numbers of encoder and decoder layers trained on a forecasting horizon of 192 on the traffic dataset. The results in Fig. 10 and Fig. 11 show that changing the number of encoder and decoder layers does not alter the trends in the profile of the ID and MAPC.

---

[1] https://www.bgc-jena.mpg.de/wetter/
[2] https://archive.ics.uci.edu/dataset/321/electricityloaddiagrams20112014
[3] https://pems.dot.ca.gov/

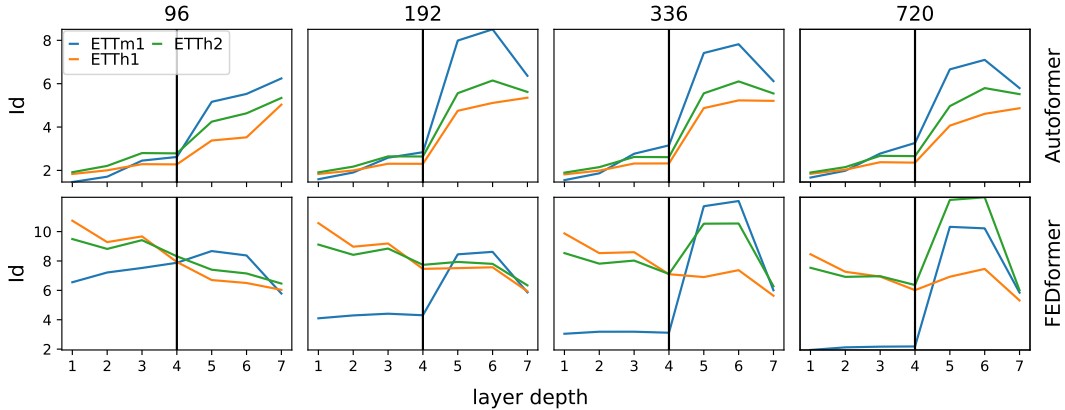

Figure 8: **ID profiles across layers of Autoformer and FEDformer on ETT datasets for multiple forecasting horizons.** Each panel includes separate ID profiles per dataset, for several horizons (left to right) and architectures (top to bottom).

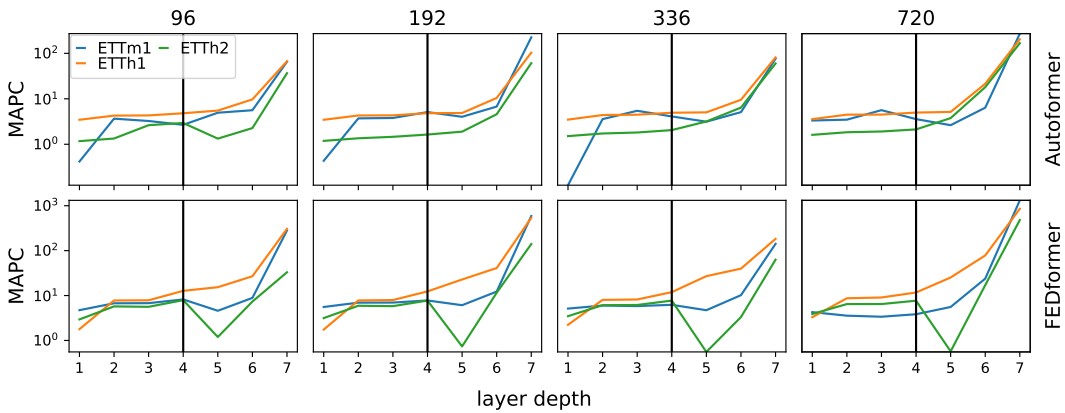

Figure 9: **MAPC profiles across layers of Autoformer and FEDformer on ETT datasets for multiple forecasting horizons.** Each panel includes separate MAPC profiles per dataset, for several horizons (left to right) and architectures (top to bottom).

## B.3 LATENT REPRESENTATION VISUALIZATION

Our results show two geometrical properties of the latent representation for the selected layers. To further visualise the latent data, we add a two-dimensional t-SNE visualization. In Fig. 12 a clear separation appears between the encoder, decoder and the output layer. Although the two-dimensional projection does not reveal all the geometrical properties of the data, we can see that points sampled from each part in the architecture (encoder, decoder and linear layer) lie close together.

## B.4 SYNTHETIC DATASET

For further analysis we created a synthetic dataset of known intrinsic dimension and curvature. The dataset, composed of 7500 points was generated by sampling a random point $x_0 \in \mathbb{R}^3$ on the unit sphere and simulating a trajectory along the sphere. The elements of the dataset lie on the unit sphere, which has an intrinsic dimension of two and a Gaussian curvature of one. The points were embedded in a 16 dimensional space via a random orthogonal transformation which preserves the intrinsic dimension and curvature. The results in Fig. 13 show similar trend to the other tested data sets. Eventough the groundtruth ID and MAPC are known, the structure of the learned manifolds is hard to predict and it needs not to comply with the structure of the input data.

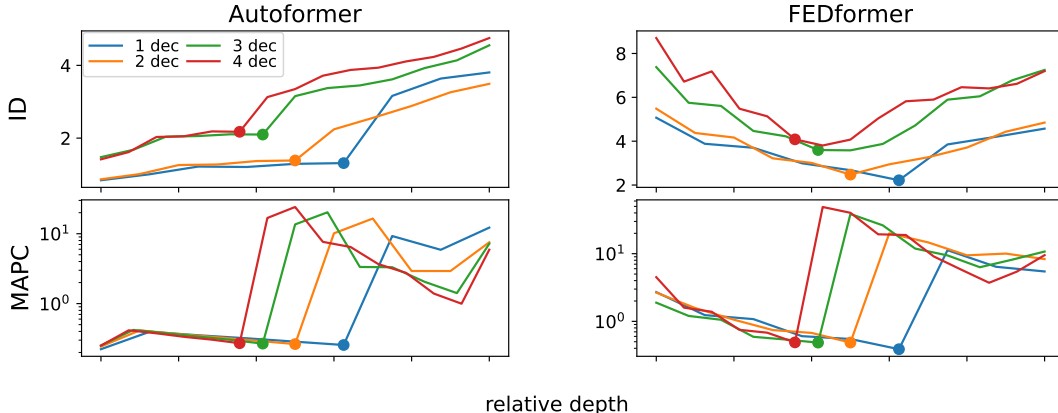

Figure 10: **Intrinsic dimension and mean absolute principal curvature along the layers of Autoformer and FEDfromer on traffic dataset with three encoder layers and varying number of decoder layers.** Top) intrinsic dimension. Bottom) mean absolute principal curvature. The large dot marks the transition from the encoders to the decoders. For each model, both ID and MAPC share a similar profile across different number of decoders.

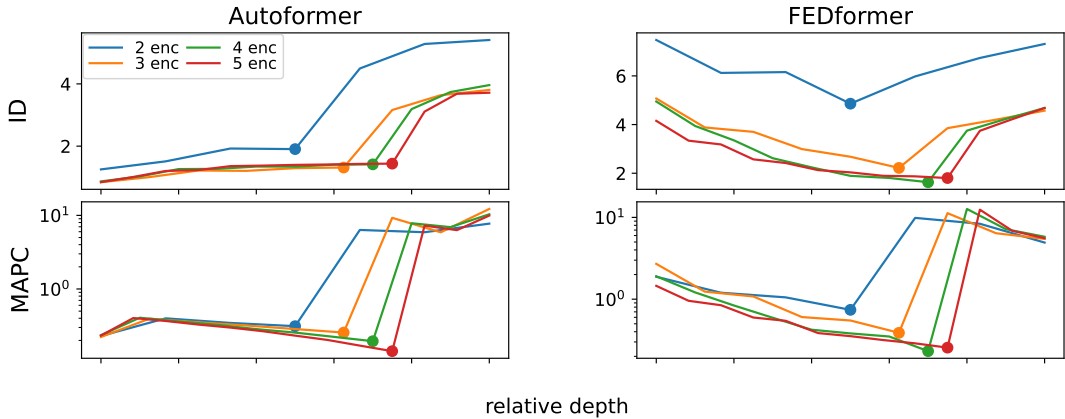

Figure 11: **Intrinsic dimension and mean absolute principal curvature along the layers of Autoformer and FEDfromer on traffic dataset with one decoder layers and varying number of encoder layers.** Top) intrinsic dimension. Bottom) mean absolute principal curvature. The large dot marks the transition from the encoders to the decoders. For each model, both ID and MAPC share a similar profile across different number of decoders.

## C  TSF MODELS

Here we will provide a supplementary analysis of additional TSF models: vanilla Transformer (Vaswani et al., 2017) and Informer (Zhou et al., 2021).

### C.1  MODEL HYPERPARAMETERS

All the Transformer-based models (Transformer, Informer, Autoformer and FEDformer) used in the paper are composed of two encoders and a single decoder, the input dimension to the tokenization layer is the dimensionality of the data (number of features), while its output (d_model - the number of expected features in the encoder/decoder inputs) is 512. The number of heads in the multihead attention layers is set to 8 and the dimension of the feedforward network model is 2048. The input data is embedded using all of its features such that the information between the features is mixed. For more information about the models and hyper parameters please head to the FEDformer repository at: `https://github.com/MAZiqing/FEDformer`

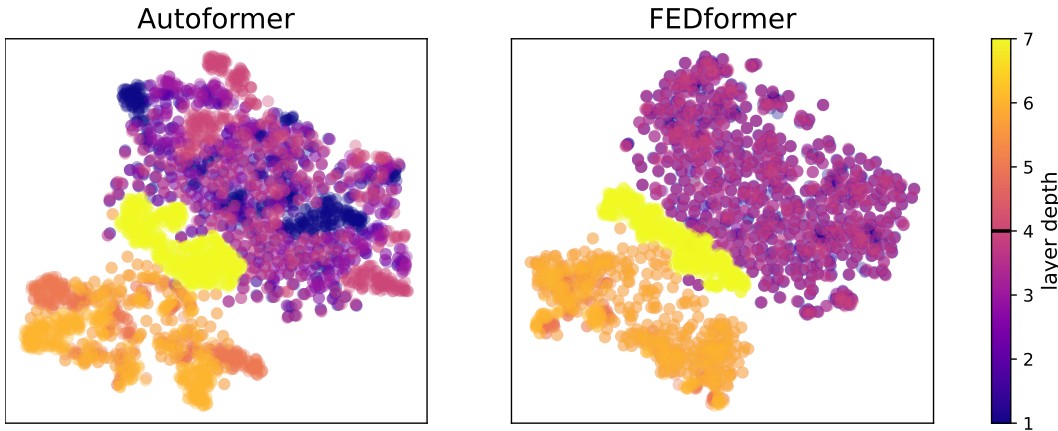

Figure 12: **t-SNE visualization of the latent representations of Autoformer and FEDfromer on ETTM1 dataset** Both architectures exhibit a separation between the encoder, decoder and the output linear layer.

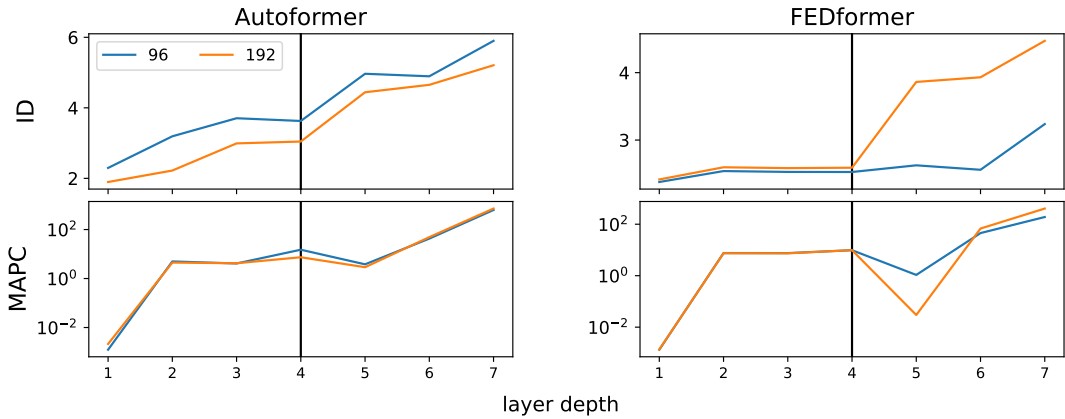

Figure 13: **Intrinsic dimension and mean absolute principal curvature along the layers of Autoformer and FEDformer on synthetic sphere dataset for multiple forecasting horizons.** Top) intrinsic dimension. Bottom) mean absolute principal curvature.

**Transformer and Informer:** These models have a similar architecture to the Autoformer and FED-former as shown in Fig. 1. The models are composed of two encoder layers and one decoder layer, however, in contrast to Autoformer and FEDformer, Transformer and Informer do not contain series decomposition layers. The analysis of Transformer and Informer inspects the output of the encoder layers, decoder layer and the last linear layer. In Fig. 14 we observe trends similar to ones shown for the Autoformer and FEDformer. When comparing different datasets, we see similar trends, a monotonic increase in ID for the Transformer and a saw-like behaviour for Informer (see Fig.15). The MAPC across datasets for the Transformer exhibits a monotonic increase trend while the Informer has a "v"-shape (see Fig.16)

# D   INTRINSIC DIMENSION AND MEAN ABSOLUTE PRINCIPAL CURVATURE

## D.1   RELIABILITY OF TWONN AND MAPC IN HIGH-DIMENSIONS

We would like to note that both TwoNN and CAML were proven reliable for high dimensions when introduced in their respective original papers (Facco et al., 2017; Li, 2018), as well as in analysis works on NN (Ansuini et al., 2019; Kaufman & Azencot, 2023). Specifically, TwoNN was shown to be reliable if the ID is smaller than 20 (Facco et al., 2017; Ansuini et al., 2019), even for high-dimensional inputs. Similarly, CAML was shown to be robust for high-dimensional data on

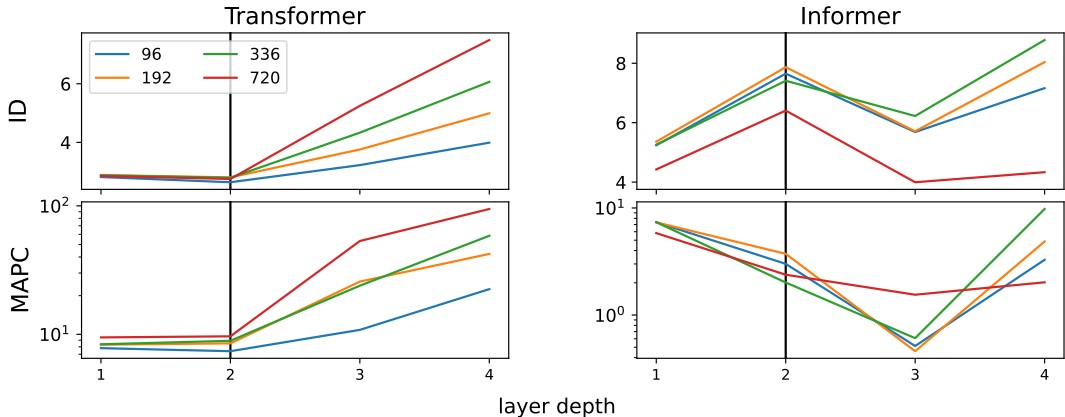

Figure 14: **Intrinsic dimension and mean absolute principal curvature along the layers of Transformer and Informer on traffic dataset for multiple forecasting horizons.** Top) intrinsic dimension. Bottom) mean absolute principal curvature. For each model, both ID and MAPC share a similar profile across different forecasting horizons.

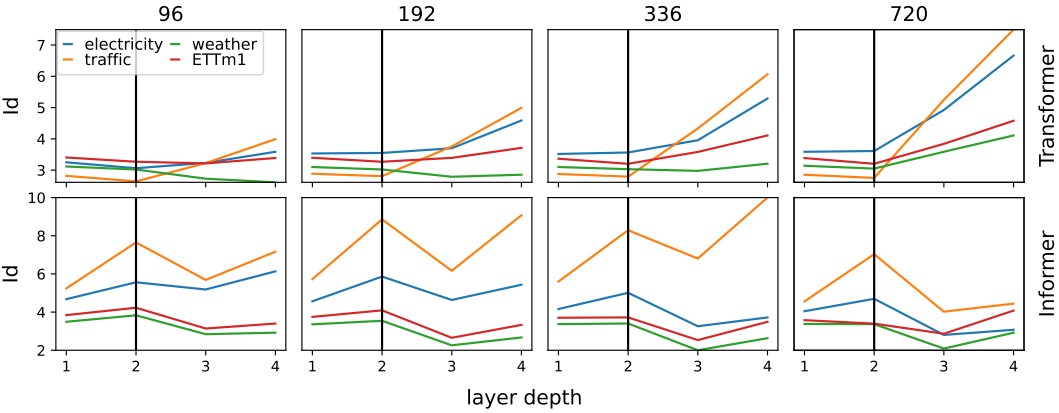

Figure 15: **MAPC profiles across layers of Transformer and Informer on electricity, traffic, weather and ETTm1 datasets for multiple forecasting horizons.** Each panel includes separate MAPC profiles per dataset, for several horizons (left to right) and architectures (top to bottom).

several manifolds with known curvature (Li, 2018; Kaufman & Azencot, 2023). In addition, we tested TwoNN and CAML for reliability on a synthetic dataset with known intrinsic dimension and curvature. We sampled 1000 points from a sphere with radius 4 and then embedded them into a 10000 dimensional space. In this test, the extrinsic dimension of latent representations does not exceed 2048. The estimated id was 1.96, where the ground truth intrinsic dimension is 2. Further, the curvature was 0.118 with a ground truth Gaussian curvature of 0.111.

## D.2 INTRINSIC DIMENSION

To estimate the ID of data representations in TSF neural networks, we use the TwoNN (Facco et al., 2017) global id estimator. The ID-estimator utilizes the distances only to the first two nearest neighbors of each point. This minimal selection helps reduce the impact of inconsistencies in the dataset during the estimation process.

**Method** Let $X = \{x_1, x_2, \cdots, x_N\}$ a set of points uniformly sampled on a manifold with intrinsic dimension $d$. For each point $x_i$, we find the two shortest distances $r_1, r_2$ from elements in $X \setminus \{x_i\}$ and compute the ratio $\mu_i = \frac{r_2}{r_1}$. It can be shown that $\mu_i, 1 \le i \le N$ follow a Pareto distribution with parameter $d+1$ on $[1, \infty)$, that is $f(\mu_i \mid d) = d\mu_i^{-(d+1)}$. While $d$ can be estimated by maximizing

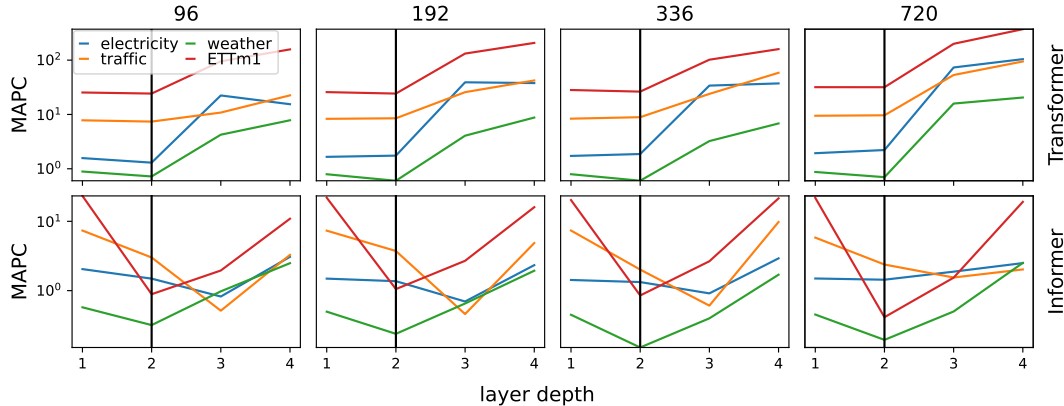

Figure 16: **MAPC profiles across layers of v on electricity, traffic, weather and ETTm1 datasets for multiple forecasting horizons.** Each panel includes separate MAPC profiles per dataset, for several horizons (left to right) and architectures (top to bottom).

the likelihood:

$$P(\mu_1, \mu_2, \cdots \mu_N \mid d) = d^N \prod_{i=1}^{N} \mu_i^{-(d+1)} \quad (1)$$

we follow the method proposed by (Facco et al., 2017) based on the cumulative distribution $F(\mu) = 1 - \mu^{-d}$. The idea is to estimate $d$ by a linear regression on the empirical estimate of $F(\mu)$. This is done by sorting the values of $\mu$ in ascending order and defining $F^{emp}(\mu_i) \doteq \frac{i}{N}$. A straight line is then fitted on the points of the plane $\{(\log \mu_i, -\log(1 - F_i^{emp}))\}_{i=1}^{N}$. The slope of the line is the estimated ID.

### D.3 DATA DENSITY

In comparison to the intrinsic dimension, which is a characteristic of the entire manifold, curvature information is local. Moreover, curvatures are calculated using second-order derivatives of the manifold. Consequently, our study assumes that the data is dense enough to compute curvatures. However, the latent representations of data, are both high-dimensional and sparse, which presents significant difficulties in calculating local differentiable values on such as curvature.

The typical characteristics of data used in machine learning require a large number of nearby points to create a stable neighborhood. One commonly used tool for this is k-Nearest-Neighbours (KNN). However, KNN can sometimes generate non-local and sparse neighborhoods, where the "neighbors" are effectively far apart in a Euclidean sense. Another approach is to use domain-specific augmentations, such as window cropping, window warping or slicing. However, this approach only explores a specific aspect of the data manifold and may overlook other important parts. A more effective approach, regardless of the domain, is to compute the Singular Value Decomposition (SVD) for each time series. This generates a close neighborhood by filtering out small amounts of noise in the data. This approach is well-motivated from a differential geometry standpoint, as it approximates the manifold at a point and samples the neighborhood.

**Neighborhood generation.** To improve the local density of time series samples, we use a procedure similar to (Yu et al., 2018) to generate artificial new samples by reducing the "noise" levels of the original data. Specifically, given a $d$ dimensional time series $x_{1:T} \in \mathbb{R}^{T \times d}$, let $x_{1:T} = U\Sigma V^T$ be its SVD, where $U \in \mathbb{R}^{T \times T}$, $V \in \mathbb{R}^{d \times d}$ and $\Sigma \in \mathbb{R}^{T \times d}$ a rectangular diagonal matrix with singular values $\{\sigma_1, \sigma_2, \cdots, \sigma_d\}$ on the diagonal in descending order such that $r$ is the rank of $x_{1:T}$. Let $m$ be the smallest index such that the explained variance $\frac{\sigma_m^2}{\sum_j \sigma_j^2}$ of the $m$-th mode is less than or equal to $1e^{-3}$. We define $\Sigma' = \{\sigma_1, \sigma_2, \cdots, u_1\sigma_m, u_2\sigma_{m+1}, \cdots u_{d-m+1}\sigma_d\}$ such that $u_i \overset{\text{i.i.d.}}{\sim} \mathbb{U}(0, 1)$. this process is repeated $64$ times for each time series, generating $64$ new time series.

### D.4 CURVATURE ESTIMATION

There are several methods available for estimating curvature quantities of data representations, as discussed in papers such as (Brahma et al., 2015; Shao et al., 2018). For our purposes, we have chosen to use the algorithm described in (Li, 2018), which is called Curvature Aware Manifold Learning (CAML). We opted for this algorithm because it is supported by theoretical foundations and is relatively efficient. In order to use CAML, we need to provide the neighborhood information of a sample and an estimate of the unknown ID. The ID is estimated using the TwoNN algorithm, as described inD.3, similarly to (Ansuini et al., 2019; Kaufman & Azencot, 2023).

In order to estimate the curvature of data $Y = \{y_1, y_2, \cdots, y_N\} \subset \mathbb{R}^D$, we make the assumption that the data lies on a $d$-dimensional manifold $\mathcal{M}$ embedded in $\mathbb{R}^D$, where $d$ is much smaller than $D$. Consequently, $\mathcal{M}$ can be considered as a sub-manifold of $\mathbb{R}^D$. The main concept behind CAML is to compute a local approximation of the embedding map using second-order information.

$$f : \mathbb{R}^d \to \mathbb{R}^D \quad , y_i = f(x_i) + \epsilon_i , \quad i = 1, \ldots, N , \tag{2}$$

where $X = \{x_1, x_2, \cdots, x_N\} \subset \mathbb{R}^d$ are low-dimensional representations of $Y$, and $\{\epsilon_1, \epsilon_2, \cdots \epsilon_N\}$ are the noises. In the context of this paper, the embedding map $f$ is the transformation that maps the low-dimensional dynamics to the sampled features for each time stamp $t$ that might hold redundant information.

In order to estimate curvature information at a point $y_i \in Y$, we follow the procedure described above to define its neighborhood. This results in a set of nearby points $\{y_{i_1}, \ldots, y_{i_K}\}$, where $K$ represents the number of neighbors. Using this set along with the point $y_i$, we utilize SVD to construct a local natural orthonormal coordinate frame $\left\{ \frac{\partial}{\partial x^1}, \cdots, \frac{\partial}{\partial x^d}, \frac{\partial}{\partial y^1}, \cdots, \frac{\partial}{\partial y^{D-d}} \right\}$. This coordinate frame consists of a basis for the tangent space (first $d$ elements) and a basis for the normal space. To be precise, we denote the projection of $y_i$ and $y_{i_j}$ for $j = 1, \ldots, K$ onto the tangent space spanned by $\partial/\partial x^1, \ldots, \partial/\partial x^d$ as $x_i$ and $u_{i_j}$ respectively. It is important to note that the neighborhood of $y_i$ must have a rank of $r > d$. If the rank is less than $d$, then SVD cannot accurately encode the normal component at $x_i$, leading to poor approximations of $f$ at $x_i$. Therefore, we verify that $\{y_{i_1}, \ldots, y_{i_K}\}$ has a rank of $d + 1$ or higher.

The map $f$ can be expressed in the alternative coordinate frame as $f(x^1, \ldots, x^d) = [x^1, \ldots, x^d, f^1, \ldots, f^{D-d}]$. The second-order Taylor expansion of $f^\alpha$ at $u_{i_j}$ with respect to $x_i$, with an error of $\mathcal{O}(|u_{i_j}|_2^2)$, is represented by

$$f^\alpha(u_{i_j}) \approx f^\alpha(x_i) + \Delta_{x_i}^T \nabla f^\alpha + \frac{1}{2} \Delta_{x_i}^T H^\alpha \Delta_{x_i} , \tag{3}$$

where $\alpha = 1, \ldots, D - d$, $\Delta_{x_i} = (u_{i_j} - x_i)$ and $u_{i_j}$ is an element in the neighborhood of $x_i$. The gradient of $f^\alpha$ is denoted by $\nabla f^\alpha$, and $H^\alpha = \left( \frac{\partial^2 f^\alpha}{\partial x^i \partial x^j} \right)$ is its Hessian. We have a neighborhood $\{y_{i_1}, \ldots, y_{i_K}\}$ of $y_i$, and their corresponding tangent representations $\{u_{i_j}\}$. Using equation3, we can form a system of linear equations, as explained in D.5. The principal curvatures are the eigenvalues of $H^\alpha$, so estimating curvature information involves solving a linear regression problem followed by an eigendecomposition. Each Hessian has $d$ eigenvalues, so each sample will have $(D - d) \times d$ principal curvatures. Additionally, one can compute the Riemannian curvature tensor using the principal curvatures, but this requires high computational resources due to its large number of elements. Moreover, as the Riemannian curvature tensor is fully determined by the principal curvatures, we focus our analysis on the eigenvalues of the Hessian. To evaluate the curvature of manifolds, we estimate the mean absolute principal curvature (MAPC) by taking the mean of the absolute values of the eigenvalues of the estimated Hessian matrices.

### D.5 ESTIMATING THE HESSIAN MATRIX

In order to estimate the Hessian of the embedding mapping $f^\alpha$ where $\alpha = 1, \ldots, D - d$, we build a set of linear equations that solves Eq. 3. We approximate $f^\alpha$ by solving the system $f^\alpha = \Psi X_i$, where $X_i$ holds the unknown elements of the gradient $\nabla f^\alpha$ and the hessian $H^\alpha$. We define $f^\alpha = [f^\alpha(u_{i_1}), \cdots, f^\alpha(u_{i_K})]^T$, where $u_{i_j}$ are points in the neighborhood of $x_i$, in the local natural orthogonal coordinates. The local natural orthogonal coordinates are a set of coordinates that are

defined at a specific point $p$ of the manifold. They are constructed by finding a basis for the tangent space and normal space at a point $p$ by applying Principal Component Analysis, such that the first $d$ coordinates (associated with the most significant modes, i.e., largest singular values) represent the tangent space, and the rest represent the normal space. We define $\Psi = [\Psi_{i_1}, \cdots, \Psi_{i_K}]$, where $\Psi_{i_j}$ is given via

$$\Psi_{i_j} = \left[ u_{i_j}^1, \cdots, u_{i_j}^d, \left(u_{i_j}^1\right)^2, \cdots, \left(u_{i_j}^d\right)^2, \left(u_{i_j}^1 \times u_{i_j}^2\right), \cdots, \left(u_{i_j}^{d-1} \times u_{i_j}^d\right) \right] .$$

The set of linear equations $f^\alpha = \Psi X_i$ is solved by using the least square estimation resulting in $X_i = \Psi^\dagger f^\alpha$, where $X_i = \left[ \nabla f^{\alpha 1}, \cdots, \nabla f^{\alpha d}, H^{\alpha 1,1}, \cdots, H^{\alpha d,d}, H^{\alpha 1,2}, \cdots, H^{\alpha d-1,d} \right]$. In practice, we estimate only the upper triangular part of $H^\alpha$ since it is a symmetric matrix. The gradient values $\nabla f^\alpha$ are ignored since they are not required for the CAML algorithm. We refer the reader for a more comprehensive and detailed analysis in (Li, 2018).

