# OpenReview forum: "Analyzing Deep Transformer Models for Time Series Forecasting via Manifold Learning"
_ICLR.cc/2024/Conference — Submitted to ICLR 2024_

### Official Review · Reviewer_SZbM · 2023-10-30

**Soundness:** 3 good
**Presentation:** 3 good
**Contribution:** 3 good
**Rating:** 6
**Confidence:** 4

**Summary:**

This work analyzes transformer-based timeseries forecasting (TSF) deep neural networks from a manifold learning perspective. Specifically, the authors analyzes the behaviour of a few transformer-based models for TSF task using the intrinsic dimension (ID) and mean absolute principal curvature (MAPC) from the study of geometric features of Riemannian manifolds. The authors show that the studied architectures share similar ID and MAPC profiles for a few different datasets. Additionally, the authors showed that the final ID is positively correlated with performance; and presented how the geometry profiles change along with the training process.

**Strengths:**

- The presented work uses the manifold learning technique to study TSF models, which is a good contribution to the field. This is because (i) the testing set might not always be available as stated by the authors; (ii) Specifically for timeseries applications, the testing set might not be representative enough to describe the model’s performance, as real-world datasets in timeseries are less generalizable and not as large-scale as image or language datasets to comprehensively evaluate the model's performance.
- Good related work summary of manifold learning analysis.

**Weaknesses:**

Major weakness:
- **Lack of insights to timeseries.** The presented work, although bridges the gap of using manifold learning analysis to study TSF models, lacks insights to timeseries datasets. Specifically, timeseries datasets have their unique properties and challenges, e.g. sampling rate of the timeseries; tokenization layer (specifically, if the token is the aggregation of timeseries across different channels, as the channel-independence property studied in [1, 2, 3]; what is the proper length of the token as mentioned in [1]). The studied group of model architecture, which is transformer, is also not representative enough for timeseries applications, as many linear models [4] and previously, CNN-based models [5] has good performance on the studied task. Overall, this has limited the contribution of the proposed work.
- **Lack of studied architectures.** The selected architectures Autoformer and Fedformer are out-of-dated and contain specific components. E.g. Fedformer uses frequency-analysis to decompose the frequency information (or seasonal trends) in timeseries. The Informer mentioned in appendix helps with this lack of architectures, yet I am not sure about the `Transformer’ mentioned in appendix. What are the specs of this transformer architecture? One important aspect of timeseries study is the spatiotemporal relationship in sequences (that are fed into the transformer); as well as the input dimension of the tokenization layer. How are those selected and how the analyzed properties differ across such choices?
- **Over-claimed contributions.** In conclusion section: “Our results indicate a fundamental difference between classification and regression models: while the former networks shrink the ID significantly to extract a meaningful representation that is amenable for linear separation, regression models behave differently” this contribution seems overclaimed. It is unclear if the behaviour (ID's correlation w performance) is unique to timeseries forecasting or if it is general to all regression/generative tasks. It is also unclear if the reason of such behaviour is due to that the task is a regression task (specifically, it is possible that all timeseries tasks e.g. classification also gives such behaviour). A related question would be that: Does that mean e.g. a generative model in vision also gives better performance when the ID is higher? This contribution should not be claimed if the authors do not fundamentally reveal/prove this correlation.

[1] Nie, Yuqi, Nam H. Nguyen, Phanwadee Sinthong, and Jayant Kalagnanam. "A time series is worth 64 words: Long-term forecasting with transformers." arXiv preprint arXiv:2211.14730 (2022).
[2] Liu, Ran, Mehdi Azabou, Max Dabagia, Jingyun Xiao, and Eva Dyer. "Seeing the forest and the tree: Building representations of both individual and collective dynamics with transformers." Advances in neural information processing systems 35 (2022): 2377-2391.
[3] Zhang, Yunhao, and Junchi Yan. "Crossformer: Transformer utilizing cross-dimension dependency for multivariate time series forecasting." In The Eleventh International Conference on Learning Representations. 2022.
[4] Zeng, Ailing, Muxi Chen, Lei Zhang, and Qiang Xu. "Are transformers effective for time series forecasting?." In Proceedings of the AAAI conference on artificial intelligence, vol. 37, no. 9, pp. 11121-11128. 2023.
[5] Yue, Zhihan, Yujing Wang, Juanyong Duan, Tianmeng Yang, Congrui Huang, Yunhai Tong, and Bixiong Xu. "Ts2vec: Towards universal representation of time series." In Proceedings of the AAAI Conference on Artificial Intelligence, vol. 36, no. 8, pp. 8980-8987. 2022.

Minor weakness:
- In the related work section, analysis of transformers: “While in general they question the effectivity of transformer for forecasting, new transformer-based approaches continue to appear (Zeng et al., 2023), consistently improving the state-of-the-art results on common forecasting benchmarks.” Zeng et al., 2023 claims that a linear-based modeling approach outperforms transformer-based modeling approaches, and does not present a transformer architecture by itself. It is unclear to me if the citation here is appropriate.
- “Namely, we discard data from the red and blue trajectories” Please refer to Figure 1 in text for this sentence for clarity.
- Can the authors break down this sentence? “We hypothesize that significantly deeper and more expressive TSF models as was in (Nie et al., 2023) and is common in classification (He et al., 2016) may yield double descent forecasting architectures (Belkin et al., 2019).”

**Questions:**

As above

---

> ### Author Response · Authors · 2023-11-18
>
> We would like to thank Reviewer SZbM for identifying the contribution in our work, the importance of our results, and for finding our related work to be well written. Additionally, we value their insightful input, including posing significant questions and providing constructive suggestions to improve the quality of our paper. Below, we address the comments of Reviewer SZbM. Given the opportunity, we will be happy to incorporate the modifications listed below into a final revision.
>
> 1. > ### ... The presented work, although bridges the gap of using manifold learning analysis to study TSF models, lacks insights to timeseries datasets.}
>
>     Our work focuses on studying the geometric properties of data manifolds arising in deep TSF models. As such, we were primarily interested in insights related to the architectures. This preference can be motivated by: i) the TwoNN and MAPC tools we employ reveal more on the architecture than on the particular data; and ii) we wanted to position our work in the global context of manifold learning approaches for deep neural networks. We would like to emphasize that our work does include a few insights related to the nature of datasets, as described by the end of page 6 where we compare between electricity and traffic vs. ETT and weather. Additionally, we also discuss the expressiveness of Autoformer and FEDformer with respect to the number of channels in time series data in page 7. However, in general, we opted for a more focused analysis and consistent observations on the geometric nature of data manifolds, and our exposition and study was designed accordingly. We believe our work is complementary to existing studies with insights on time series data [1, 2, 3], and we are happy to include these works in our discussion. In addition, we will add a discussion on the limitations of our work in providing insights on data in the final version.
>
> 2. > ### ... The studied group of model architecture, which is transformer, is also not representative enough for timeseries applications, ...}
>
>      To the best of our knowledge, the current families of deep models at the state-of-the-art on time series forecasting are transformer-based and N-BEATS-based. Please note that linear models such as [4] are not deep models and thus can not be studied with our tools. We decided to focus on the transformer family as it is more established for TSF problems than N-BEATS approaches in terms of the number of published papers. Another argument for focusing on transformer models is our aim to convey a consistent and coherent understanding and empirical results on these architectures. Mixing vastly different models under the same analysis paper would potentially be confusing in our opinion.
>
>
>
>
>
> 3. > ### Lack of studied architectures. The selected architectures Autoformer and Fedformer are out-of-dated and contain specific components. ...}
>
>     Please see our response above. In addition: There are several reasons for focusing on Autoformer and FEDformer architectures in our study. First, even though Autoformer and FEDformer were released in 2021 and 2022, respectively, they are among the best variants of the Transformer model for time series forecasting. Both models contain specific components and yet they share a similar structure which allows for a comparison.  Second, similar studies to ours on deep classification neural networks considered strong baselines such as VGG and ResNet. Following a similar reasoning, we opted for established TSF models that are still considered state-of-the-art and are commonly used in applications. Third, several derivative works are closely related to Autoformer and FEDformer due to their accurate TSF. Thus, while we can not guarantee that other transformer-based TSF models will admit similar observations as reported for Autoformer and FEDformer, it is reasonable to assume that derivative work will generalize. Finally, our results and insights extend beyond Autoformer and FEDformer and also range over Informer and Transformer, covering four architectures in total.
>
> 4. > ### ... What are the specs of this transformer architecture?}
>
>     The Transformer mentioned in the appendix is a vanilla Transformer. All the Transformer-based models used in the paper are composed of two encoders and one decoder, where a channel dependant embedding is performed across all transformer variants.

---

> > ### Comment · Reviewer_SZbM · 2023-11-20
> >
> > Thank you for your response. While I believe the response and edits to the paper helped improved the clarity, I think my major concerns remain unaddressed. Specifically, the core of my original concerns "Lack of insights to timeseries" and "Lack of studied architectures" can be concluded and rephrased as such: "Does the studied property only represent the behaviour of one specific group of model with one specific type of design, or does they generalize?" With the current evidence, I tend to believe it is the former. Also, since TSF is still an active field of research, I believe the chosen "state-of-the-art" models are not considered "well-established". This is because new transformer works (e.g. PatchTST), designed with completely different principles, outperforms such transformer models, which means that the components in the studied architectures (and thus the analyzed properties) might not be representative of the TSF task at all. Thus, based on authors' response "it is reasonable to assume that derivative work will generalize", I am curious to understand, fundamentally, what are the basic assumptions or specific model categories that would allow the authors' observations to be generalized. If the authors want to claim this specific model categories to be transformers, the authors need to include results from diverse selection of transformers with different design choices that are key to the performance improvement.

---

> ### Author Response · Authors · 2023-11-18
>
> 5. > ### ...  How are those selected and how the analyzed properties differ across such choices?}
>
>     We did not perform hyperparameter optimization and architecture design, but rather used the available architectures as they appeared in the FEDformer code repository. Specifically, the input dimension to the tokenization layer is the dimensionality of the data (number of features), while its output (d\_model) is $512$ across all models. The number of attention heads is $8$ and the dimension of the fully connected layer is $2048$. We added a specification of all the models' parameters in the appendix. If more details are needed we will gladly add them.
>
> 6. > ### Over-claimed contributions.}
>
>     We totally agree with the reviewer that our exposition and conclusions should better reflect the results we report. Thus, we propose to re-phrase our claim as follows: "Our results indicate a fundamental difference between image classification and time series forecasting models: while the former networks shrink the ID significantly to extract a meaningful representation that is amenable for linear separation, time series forecasting models behave differently.”
>
> 7. > ### ... Zeng et al., 2023 claims that a linear-based modeling approach outperforms transformer-based modeling approaches, and does not present a transformer architecture by itself. It is unclear to me if the citation here is appropriate.}
>
>     We modified this sentence in the revised version as follows: "While in general Zeng et al. (2023) question the effectivity of transformer for forecasting, new transformer-based approaches continue to appear (Nie et al., 2023), consistently improving the state-of-the-art results on common forecasting benchmarks.”
>
> 8. > ### “Namely, we discard data from the red and blue trajectories” Please refer to Figure 1 in text for this sentence for clarity.}
>
>     Following the reviewer suggestion, we added a reference to Figure 1 in the revised text.
>
> 9. > ### Can the authors break down this sentence? “We hypothesize that significantly deeper and more expressive TSF models as was in (Nie et al., 2023) and is common in classification (He et al., 2016) may yield double descent forecasting architectures (Belkin et al., 2019).”}
>
>     Yes, thank you for suggesting that. We re-phrased that sentence in the revised version as follows: "We hypothesize that current TSF models are still within the classical ML bias-variance trade-off regime (Goodfellow et al., 2016). In contrast, deep classification models (e.g., He et al., 2016) exhibit double descent effects, forming more expressive and generalizable learning algorithms. We believe that a similar phenomenon of double descent will also emerge for deeper and more expressive TSF models (Nie et al., 2023)."

---

### Official Review · Reviewer_VSrC · 2023-10-31

**Soundness:** 2 fair
**Presentation:** 3 good
**Contribution:** 2 fair
**Rating:** 5
**Confidence:** 4

**Summary:**

This paper studies the geometric properties of the latent representations of deep transformer models for time series forecasting. The authors assume that the latent representations output by each layer of the model lie on a low-dimensional manifold. They estimate the intrinsic dimension (ID) and mean absolute principal curvature (MAPC) of the manifold based on latent representations extracted from the multiple Autoformer and FEDformer models trained with different random seeds on electricity, weather, and traffic datasets. The main observations are: 1) the dimensionality and curvature of the representation manifold either drop or stay fixed in the encoder layers, but then increase significantly in the decoder layers. This phenomenon is consistent across different models, datasets, and forecast horizons; 2) The intrinsic dimension in the final layer is correlated with the model performance; 3) Compared to the untrained models, the ID and MAPC of the representation manifolds change significantly during training, but converge quickly.

**Strengths:**

- Originality: The paper is the first to study the geometric properties of the latent representations of SOTA deep models for time series forecasting. The observations are interesting and may lead to new insights about the behavior of the investigated models.
- Clarity & Quality: The background, methodology, and results are clearly presented. The paper is easy to follow.
- Significance: The paper is relevant to the ICLR community because it studies the geometric properties of the **learned representations** in SOTA deep TSF models. There are too many TSF papers that focus on the model architectures but ignore what the models have learned. This paper presents a new methodology for analyzing these models.

**Weaknesses:**

- My first concern is why Autoformer and FEDformer were chosen as the models to be analyzed. The title of the paper is "Analyzing Deep Transformer Models for Time Series Forecasting via Manifold Learning", but both Autoformer and FEDformer go far beyond the standard transformer architecture. They are not even based on the standard auto-regressive decoder. Given the complexity of these models, it is hard to tell whether the observations are specific to these models due to their design choices or generalizable to other deep transformer models. I am not convinced that the observations are generalizable because of the high complexity of the chosen models.

- Given the high dimensionality of the latent representations, I am not sure whether the estimated intrinsic dimension and curvature information are reliable. The authors should provide some justification for the reliability of the estimates. For example, why is the estimated intrinsic dimension of the first layer close to 1 for Autoformer in Figure 3? This is surprising.

- The methodology part of this work is almost identical to Kaufman & Azencot, 2023. The content in Appendix D.2 & D.3 is almost a copy of Section 3 of Kaufman & Azencot, 2023. I am not sure whether this is OK.

**Questions:**

- Given the claimed intrinsic dimension is so low, is it possible to visualize the latent representations in a 2D or 3D space? This may help us better understand the geometric properties of the latent representations.

---

> ### Author Response · Authors · 2023-11-18
>
> We express our gratitude to Reviewer VSrC for their favorable remarks regarding the novelty of our study, the clarity of our exposition, and the new methodology for analyzing learned representations. Furthermore, we appreciate their valuable contributions in posing significant questions and offering constructive suggestions for enhancing the quality of our paper. In what follows, we address the specific comments put forth by Reviewer VSrC. Given the chance, we will be happy to integrate the suggested modifications outlined below into the final revision.
>
> 1. > ### My first concern is why Autoformer and FEDformer were chosen as the models to be analyzed.}
>
>     There are several reasons for focusing on Autoformer and FEDformer architectures in our study. First, while there are a few MLP models for TSF, the majority of successful TSF approaches are based on the transformer. Thus, investigating transformer architectures provides several models to study, many of which share important design choices. Second, similar studies to ours on deep classification neural networks considered strong baselines such as VGG and ResNet. Following a similar reasoning, we opted for established TSF models that are still considered state-of-the-art and are commonly used in applications. Third, while we generally agree that Autoformer and FEDformer are relatively complex models, we do not believe they are more complex than VGG and ResNet which were studied using a similar methodology. Finally, please see also the response below for an additional reason for mainly considering Autoformer and FEDformer.
>
> 2. > ### I am not convinced that the observations are generalizable because of the high complexity of the chosen models.}
>
>     Indeed, this is a fair point and we are happy to discuss this potential limitation. However, we believe that this point actually strengthens our choice for investigating Autoformer and FEDformer. Several derivative works are closely related to Autoformer and FEDformer due to their accurate TSF. Thus, while we can not guarantee that other transformer-based TSF models will admit similar observations as reported for Autoformer and FEDformer, it is reasonable to assume that derivative work will generalize. In addition, we would like to note that our appendix also includes results for the architectures Transformer and Informer. Their geometric properties align with our findings on Autoformer and FEDformer. Therefore, our work comprises of four different transformer architectures for TSF whose geometric features agree. We believe that the robustness and consistency of our results serve as a decent baseline for future analysis of TSF architectures.
>
> 3. > ### Given the high dimensionality of the latent representations, I am not sure whether the estimated intrinsic dimension and curvature information are reliable.}
>
>     Thank you for raising this point. We would like to note that both TwoNN and CAML were proven reliable for high dimensions when introduced in their respective original papers [1, 2], as well as in analysis works on NN [3, 4]. Specifically, TwoNN was shown to be reliable if the ID is smaller than 20 [1, 3], even for high-dimensional inputs. Similarly, CAML was shown to be robust for high-dimensional data on several manifolds with known curvature [2, 4]. In addition, we tested TwoNN and CAML for reliability on a synthetic dataset with known intrinsic dimension and curvature. We sampled $1000$ points from a sphere with radius $4$ and then embedded them into a $10000$ dimensional space. In this test, the extrinsic dimension of latent representations does not exceed $2048$. The estimated id was $1.96$, where the ground truth intrinsic dimension is $2$. Further, the curvature was $0.118$ with a ground truth Gaussian curvature of $0.111$. We added this example to the revised appendix.
>
>     [1] "Estimating the intrinsic dimension of datasets by a minimal neighborhood information." by Elena Facco, et al.
>
>     [2] "Curvature-aware manifold learning." by Yangyang Li.
>
>     [3] "Intrinsic dimension of data representations in deep neural networks." by Alessio Ansuini, et al.
>
>     [4] "Data Representations' Study of Latent Image Manifolds." by Ilya Kaufman, and Omri Azencot.
>
> 4. > ### For example, why is the estimated intrinsic dimension of the first layer close to 1 for Autoformer in Figure 3? This is surprising.}
>
>     Indeed, we observe very low ID estimates on all datasets and horizons in the first layer of Autoformer as shown in Figure 3. We hypothesize that the seasonality extraction in the decomposition blocks of Autoformer is less effective and expressive in comparison to the extraction with the Frequency Enhanced Block in FEDformer.

---

> ### Author Response · Authors · 2023-11-18
>
> 5. > ### The methodology part of this work is almost identical to Kaufman \& Azencot, 2023.}
>
>     Thank you for pointing this out. We would like to note that our study employs TwoNN and MAPC, similarly to Ansuini et al. and Kaufman and Azencot. Thus, our methodology section in the appendix shares a lot in common with these works, and is provided mainly for completeness. We are happy to revise this section to highlight the important aspects of our work which are different from existing approaches in the final version.
>
> 6. > ### Given the claimed intrinsic dimension is so low, is it possible to visualize the latent representations in a 2D or 3D space?}
>
>     We will add a visualization of the latent representations in the next few days.

---

> > ### Comment · Reviewer_VSrC · 2023-11-22
> >
> > Thanks for your clarification. I would like to keep the score unchanged.

---

### Official Review · Reviewer_qBpo · 2023-11-01

**Soundness:** 2 fair
**Presentation:** 2 fair
**Contribution:** 2 fair
**Rating:** 3
**Confidence:** 3

**Summary:**

- The paper discusses the estimation of curvature quantities in data representations using the Curvature Aware Manifold Learning (CAML) algorithm. Such algorithm takes neighborhood information and an estimate of the unknown intrinsic dimension (ID) of the data.
- The paper also discusses the application of CAML to deep forecasting models and analyzes the intrinsic dimension and mean absolute principal curvature profiles across layers.
- The analysis is conducted on various datasets and architectures, providing insights into the geometric features of deep forecasting models.

----
Thanks for the rebuttal, but I would remain the same score.

**Strengths:**

- The paper explores the estimation of curvature quantities in data representations using the Curvature Aware Manifold Learning (CAML) algorithm.
- The paper investigates the geometric properties of data manifolds across layers and provides insights into the intrinsic dimension (ID) and mean absolute principal curvature (MAPC) profiles.
- The correlation between the intrinsic dimension in the final layer and model performance is explored, providing valuable insights for model evaluation and comparison.

**Weaknesses:**

- Theoretical contribution is limited in this paper. ID is computed by other paper (TwoNN method), and MAPC ( we employ the curvature aware manifold learning (CAML) technique). Thus, both important tools of this paper is borrowed from other paper.
- Experimentally, the paper only briefly mentions the potential reasons for the observed behavior of the models but does not delve into a thorough discussion of the implications and practical implications of the findings. This could limit the broader understanding and application of the research.

**Questions:**

- What is the main theoretical contribution of this paper, if exists.
- What is the main take-home message of the findings from this paper and why it is important.

---

> ### Author Response · Authors · 2023-11-18
>
> We are thankful to Reviewer qBpo for recognizing the insights presented in our work, particularly the correlation between ID and model performance. Below, we respond to the specific comments provided by Reviewer qBpo. If given the opportunity, we would be pleased to incorporate the proposed modifications outlined below into the final revision.
>
> 1. > ### Theoretical contribution is limited in this paper. ...}
>
>     Our work characterizes the geometric properties of latent data manifolds arising in time series forecasting transformer models. While our study is primarily empirical, our results and analysis have theoretical contributions. First, we suggest a qualitative characterization of latent manifolds in TSF where geometric profiles follow an encoding phase with decreasing or fixed ID and MAPC and a decoding phase with increasing geometric measures. Second, we position our analysis in the wider context of literature that aims to characterize the geometric features of deep neural networks. Third, we propose a new tool for comparing TSF models without accessing the test set. To the best of our knowledge, we are first to suggest such a tool for TSF tasks. Finally, we show that untrained TSF models exhibit a different behavior, and thus, the ID and MAPC profiles found after training are a unique characteristic of the learning process.
>
> 2. > ### Experimentally, the paper only briefly mentions the potential reasons for the observed behavior of the models but does not delve into a thorough discussion of the implications and practical implications of the findings. ...}
>
>     We do not feel that our work "only briefly mentions the potential reasons for the observed behavior ...". For instance, one of our main claims regarding the potential reasons for the observed behavior are related to the nature of the problem (TSF) and the architecture design (encoder-decoder models). Specifically, we observe that inputs and outputs of TSF models are of the same domain, and thus, should exhibit similar geometric profiles. Then, we justify the geometric measures in inner layers by suggesting that TSF models and particularly encoder-decoder frameworks learn simpler latent representation to successfully solve the problem. Additionally, every result in the main paper is accompanied by text that describes the results, analyzes them, and suggests potential reasons related to the underlying problem, model or data.
>
>     We agree with the reviewer that our work does not focus on practical implications. We do suggest a new tool for comparing models which has several straightforward implications. However, in general, we aimed for a document that characterizes the latent data manifolds from a manifold learning perspective, and we leave the practical implications of our observations for future work. Similar works as ours took a similar approach (Ansuini et al., 2019, Valeriani et al., 2023, Kaufman et al., 2023).
>
> 3. > ### What is the main theoretical contribution of this paper , if exists.}
>
>     Please see our response above.
>
> 4. > ### What is the main take-home message of the findings from this paper and why it is important.}
>
>     We show several important findings: deep transformer TSF models share a similar geometric behavior across layers, and that geometric features are correlated with model performance. Further, untrained models present different structures, which rapidly converge during training. Our geometric analysis may be used in designing new and improved deep forecasting neural nets. For instance, one may design models whose ID or MAPC follow a characteristic profile similar to what we found. There could be various ways to achieve this by e.g., parametrizing the learned representations to a certain distribution, or by e.g., penalizing unfavorable representation through the objective function. We leave further consideration and exploration of this direction to future work.

---

### Official Review · Reviewer_xeJH · 2023-11-05

**Soundness:** 3 good
**Presentation:** 2 fair
**Contribution:** 3 good
**Rating:** 5
**Confidence:** 4

**Summary:**

In this work the authors explore the use of manifold learning to attempt to illuminate the training and performance of transformer-based deep time-series forecasting. They utilize intrinsic dimension and mean absolute principal curvature to gain insights into behaviors across layers and training.

**Strengths:**

The authors take an interesting and somewhat novel (in terms of application task) to studying the behavior of transformer-based models for time-series forecasting.

The reveal an interesting and consistent distribution of the principal curvatures across datasets and highlight a distinction between the behavior of transformer models for classification and regression.

Although difficult to decipher the connections at times, each of the core claims made at the end of the introduction are supported by the experimental results.

**Weaknesses:**

The first claimed result in the introduction is "during encoding, dimensionality and curvature either drop or stay fixed, and then, during
the decoding part, both dimensionality and curvature increase significantly." This is presumably supported in figures 2 and 3. However, there is no really compelling definition of "significant." The FEDFormer model, for example, ID does not appear to increase significantly but rather end at slightly lower values than at the initial layer. Figure 3 would benefit from the inclusion of errorbars.

While it may be a gap in the reviewer's understanding, the statement that "Indeed, regression models as TSF are expected to learn an underlying low-dimensional and simple representation while encoding. Then, a more complex manifold that better reflects the properties of the input data is learned during decoding" is not intuitively obvious from the results. Seemingly, the inverse relationship between MSE and ID suggests more faithful predictions on the test set but that does not immediately mean that it is capturing properties of the input data. For example, if the trends held for a synthetic time-series dataset based on a dynamical system representing a known manifold structure or where a ground-truth curvature was known, it would more strongly convince this reviewer.

While the reviewer appreciates that the authors still included ETTm1 in their plots, it does exhibit visually distinct behaviors that don't feel convincingly described away by stating that it is because it has fewer features. It would be helpful for a more comprehensive explanation of this either in the text or point to a discussion in the appendix.

Other comments:
Including a vertical bar identifying the shift from encoder to decoder layers would make it more clear where the authors are identifying the trends that are intended to support the claims.

Adding the correlation coefficient to Figure 5 would be appreciated.

**Questions:**

The authors state "Indeed, regression models asTSF are expected to learn an underlying low-dimensional and simple representation while encoding." Why is this what one would expect?

Would one expect this behavior to hold for models with more layers? Why?

It appears that the distributions of constant curvatures are converging. Do you have a hypothesis for what it is converging to and why (for each of the encoder and decoders)?

Did you consider, or could you add, results from a synthetic, well-understood dataset?

---

> ### Author Response · Authors · 2023-11-12
> **Clarification**
>
> Thank you for your detailed review. Before we address all the remarks, a clarification would be much appreciated.
> Could you explain what you mean by “constant curvatures are converging”? Do you refer to the convergence of MAPC values of the training dynamics in figure 6 or the convergence of principal curvatures in the deeper layers in figure 7?

---

> > ### Comment · Reviewer_xeJH · 2023-11-22
> >
> > I was referring to Figure 7. In the plots it appears that the different curves are converging towards a common shape/distribution. Do you have an intuition of what this distribution is and why?

---

> ### Author Response · Authors · 2023-11-18
>
> We express our gratitude to Reviewer xeJH for their favorable remarks regarding the novelty of our work, the empirical characterization of curvature, and that the core claims are supported by experimental results. Furthermore, we appreciate their valuable contributions in posing significant questions and offering constructive suggestions for enhancing the quality of our paper. In what follows, we address the specific comments put forth by Reviewer xeJH. Given the chance, we will be happy to integrate the suggested modifications outlined below into the final revision.
>
> 1. > ### ... This is presumably supported in figures 2 and 3. However, there is no really compelling definition of "significant." The FEDFormer model, for example, ID does not appear to increase significantly but rather end at slightly lower values than at the initial layer.}
>
>     Our claim regarding the encoding and decoding should be re-phrased for clarification. Essentially, the text should reflect the distinction between the encoding phase and the decoding phase. During encoding, dimensionality and curvature either drop or stay relatively fixed, whereas, during decoding, the dimensionality and curvature increase with respect to the initial dimensionality and curvature in the decoder (and not with respect to their initial values). We updated the text accordingly in our revised version.
>
>
> 2. > ### Figure 3 would benefit from the inclusion of errorbars.}
>
>     Following the reviewer suggestion, we added error bars to Figure 3 in the revised version.
>
> 3. > ### ... "Indeed, regression models as TSF are expected to learn an underlying low-dimensional and simple representation while encoding. Then, a more complex manifold that better reflects the properties of the input data is learned during decoding" is not intuitively obvious from the results.}
>
>     The discussion referred by the reviewer is on one of the main differences between classification and time series forecasting. A classification model is typically viewed as transforming the inputs to a latent space where linear classification becomes easy [1]. In contrast, time series forecasting is a different problem, where the inputs and outputs (i.e., forecasts) share the same domain. Under the typical assumption that the input space is complex, it is natural to assume that TSF models learn a simpler representation of the data during the encoding phase, from which it is easier to derive accurate forecasts. In our discussion and analysis of results, we find a simplifying pattern during the encoding phase with respect to ID and curvature, followed by learning complex structures during the decoding phase. We clarified and extended our discussion in the revised version.
>
>     [1] "Deep Learning." by Ian Goodfellow, Yoshua Bengio, and Aaron Courville
>
> 4. > ### For example, if the trends held for a synthetic time-series dataset based on a dynamical system representing a known manifold structure or where a ground-truth curvature was known, it would more strongly convince this reviewer.}
>
> We will add results on a synthetic time-series dataset in the next few days.
>
> 5. > ### ... ETTm1 in their plots, it does exhibit visually distinct behaviors that don't feel convincingly described away by stating that it is because it has fewer features. It would be helpful for a more comprehensive explanation of this either in the text or point to a discussion in the appendix.}
>
>     Thank you for giving us the opportunity to further discuss this phenomenon. Based on Figure 3, ETTm1 presents similar ID profiles to electricity and traffic in Autoformer and weather in FEDformer. The main qualitative difference between the ID of ETTm1 and the other datasets is the typical drop in ID in the last layer. This phenomenon can be fully explained by the number of features in ETTm1: the output is of dimension seven, and thus its ID can not be larger than that number. The MAPC results in Figure 4 show that ETTm1 follows a similar decoding pattern as weather on both Autoformer and FEDformer. However, in Autoformer, ETTm1 exhibits an increase in curvature early on in the encoder. Combining this finding with Figure 9, we see that other variants of ETT show high MAPC values in Autoformer from the beginning of the encoding phase. A similar behavior appears for the rest of the datasets that present a relatively constant MAPC during encoding. We will add this discussion in the revised appendix and refer to it from the main text.
>
> 6. > ### Including a vertical bar identifying the shift from encoder to decoder layers would make it more clear where the authors are identifying the trends that are intended to support the claims.}
>
>     Thank you for suggesting that! We added vertical bars between the encoder and the decoder to the figures in the revised version.

---

> > ### Comment · Reviewer_xeJH · 2023-11-22
> >
> > Thank you for your responses and for making some of the changes recommended for clarity. Had some of the additional results (around synthetic data) been included, it might have strengthened the claims. The reviewer sees the changes as having improved the paper but not sufficiently so to change the score

---

> ### Author Response · Authors · 2023-11-18
>
> 7. > ### Adding the correlation coefficient to Figure 5 would be appreciated.}
>
>     We will add the correlation coefficient in the revised version.
>
> 8. > ### The authors state "Indeed, regression models as TSF are expected to learn an underlying low-dimensional and simple
>     representation while encoding." Why is this what one would expect?}
>
>     Please see our response above.
>
> 9. > ### Would one expect this behavior to hold for models with more layers? Why?}
>
>     A common assumption on learning models is that depth improves the learned latent representation. While this assumption is not immediately related to geometric properties, prior work on classification does show consistent ID profiles with respect to the relative depth [1]. Namely, increasing the number of layers does not change the qualitative behavior.  The revised version includes an analysis of the Autoformer and FEDfromer models with varying numbers of encoder and decoder layers trained on a forecasting horizon of 192 on the traffic dataset. The results show that changing the number of encoder and decoder layers does not alter the trends in the profile of the ID and MAPC.
>
>     [1] "Intrinsic dimension of data representations in deep neural networks." by Alessio Ansuini, et al.
>
> 10. > ### It appears that the distributions of constant curvatures are converging. Do you have a hypothesis for what it is converging to and why (for each of the encoder and decoders)?}
>
>     We asked for a clarification on this question, but unfortunately. We did not receive one yet. In our response, we assume the reviewer refers to Figure 6, where the curvatures of the encoder phase converge. Our main hypothesis for this behavior is that the encoder is potentially not improving its initial manifold. This may be due to training is stuck in a local minima or a non-expressive encoder module. Extending and generalizing this experiment for the training dynamics is intersting, and will be the focus of future work.
>
> 11. > ### Did you consider , or could you add, results from a synthetic, well-understood dataset?}
>
>     Please see our response above.

---

### Meta-Review · Area_Chair_9nh8 · 2023-12-07

**Metareview:**

This paper presents a study of intrinsic dimensionality (ID) of features at different layers of transformer-based time-series forecasting models, with previously proposed methods for  ID estimation. The authors show a somewhat consistent trend of ID distribution for the architectures explored. However, the reviewers are concerned that the paper does not offer enough insight regarding time series data, e.g., what unique structure of time series caused such phenomena, do the results generalize to other neural architectures, and do they help develop improved models.

**Justification For Why Not Higher Score:**

Most reviewers are concerned that the paper does not have sufficient technical development, and has questions on impact and generalization of results. While reviewer SZbM gave the highest rating 6, his/her reviewer was quite critical and the rebuttal did not change his/her mind.

**Justification For Why Not Lower Score:**

N/A

---

### Decision · Program_Chairs · 2024-01-16

Reject